# Structural basis for GPR40 allosteric agonism and incretin stimulation

Joseph D. Ho[1], Betty Chau[1], Logan Rodgers[1], Frances Lu[1], Kelly L. Wilbur[2], Keith A. Otto[2], Yanyun Chen[2], Min Song[2], Jonathan P. Riley[2], Hsiu-Chiung Yang[2], Nichole A. Reynolds[2], Steven D. Kahl[2], Anjana Patel Lewis[2], Christopher Groshong[1], Russell E. Madsen[1], Kris Conners[1], Jayana P. Lineswala[2], Tarun Gheyi[1], Melbert-Brian Decipulo Saflor[1], Matthew R. Lee[1], Jordi Benach[3], Kenton A. Baker[1], Chahrzad Montrose-Rafizadeh[2], Michael J. Genin[2], Anne R. Miller[2] & Chafiq Hamdouchi[2]

Activation of free fatty acid receptor 1 (GPR40) by synthetic partial and full agonists occur via distinct allosteric sites. A crystal structure of GPR40-TAK-875 complex revealed the allosteric site for the partial agonist. Here we report the 2.76-Å crystal structure of human GPR40 in complex with a synthetic full agonist, compound **1**, bound to the second allosteric site. Unlike TAK-875, which acts as a $G\alpha_q$-coupled partial agonist, compound **1** is a dual $G\alpha_q$ and $G\alpha_s$-coupled full agonist. compound **1** binds in the lipid-rich region of the receptor near intracellular loop 2 (ICL2), in which the stabilization of ICL2 by the ligand is likely the primary mechanism for the enhanced G protein activities. The endogenous free fatty acid (FFA), γ-linolenic acid, can be computationally modeled in this site. Both γ-linolenic acid and compound **1** exhibit positive cooperativity with TAK-875, suggesting that this site could also serve as a FFA binding site.

---

[1] Lilly Biotechnology Center San Diego, 10290 Campus Point Drive, San Diego, CA 92121, USA. [2] Lilly Research Laboratories, Lilly Corporate Center, 355 East Merrill Street, Indianapolis, IN 46285, USA. [3] Lilly Research Laboratories Collaborative Access Team, Advanced Photon Source, Argonne National Laboratory, Building 438A, 9700 S. Cass Avenue, Lemont, IL 60439, USA. Correspondence and requests for materials should be addressed to J.D.H. (email: ho_joseph_d@lilly.com) or to C.H. (email: hamdouchi_chafiq@lilly.com)

World Health Organization reported in 2014 that diabetes is affecting an estimated 422 million adults globally, a nearly four-fold increase from 108 million cases in 1980[1]. In the United States about 1.7 million new cases of diabetes are reported each year—a rate that could lead to one out of every three adults having diabetes by 2050[2]. Type 2 diabetes mellitus (T2DM) accounts for ~ 90% of all diabetes[3], and the basis for treating T2DM is improvement of glycemic control. While numerous clinical treatments of T2DM are available today, many are associated with negative side effects such as hypoglycemia and weight gain; thus, there remains a demand for new anti-diabetic medicines with improved metabolic profiles. In the last decade, free fatty acid receptor 1 (FFAR1 or GPR40) has emerged as an attractive diabetes therapeutic target with both glucose-lowering and weight loss potential[4]. A G-protein coupled receptor that responds to dietary long-chain free fatty acids (FFAs)[5], GPR40 modulates FFA-stimulated insulin secretion in pancreatic β cells and incretin secretion in enteroendocrine cells in a glucose dependent manner[6–9]. Among the clinically studied synthetic GPR40 agonists, TAK-875 advanced to phase III clinical trial before its termination due to toxicity[10,11]. While clinical data demonstrated that TAK-875 has potent anti-diabetic effects with minimal incidence of hypoglycemia and weight gain[12], in vitro and in vivo studies showed that TAK-875 functioned as a partial agonist and minimally effected incretin release from enteroendocrine cells[13,14]. Meanwhile, AM-1638 was discovered among a novel series of agonists that exerts full agonism on GPR40 and stimulates incretin secretion[15]. This class of full agonists also acts allosterically with the endogenous FFAs, but binds to a topographically distinct site from TAK-875[16]. Moreover, unlike TAK-875 which activates only the $G\alpha_q/Ca^{2+}$ pathway, these full agonists allow GPR40 to couple to both the $G\alpha_s/$ cAMP and $G\alpha_q/Ca^{2+}$ signaling pathways, potentially explaining their unique activity as potent incretin secretagogues[14]. Based upon the disparate pharmacology of these compounds, a widely held hypothesis suggests that aside from the orthosteric site for engaging endogenous FFAs, GPR40 has an additional two distinct allosteric binding sites: A1 engages ligands such as TAK-875 and A2 engages ligands such as AM-1638[10]. In 2014, the crystal structure of GPR40 bound to TAK-875 elucidated the location of the first allosteric site (A1)[17]. Here we report the 2.76-Å crystal structure of human GPR40 complexed with compound **1** bound in a second structurally distinct allosteric site (A2), located at the receptor side facing the membrane lipophilic environment. Binding of compound **1** stabilizes the intracellular loop 2 (ICL2) of GPR40 in a helical conformation. Mutagenesis studies in ICL2 shows that this loop is important for $G\alpha_s$ coupling. Furthermore, positive functional cooperativity is observed between TAK-875 and compound **1** as the reported TAK-875 activity augmentation by γ-linolenic acid (γ-LA)[13]. γ-LA can be modeled into the A2 site using docking and free energy calculations, raising the possibility that site A2 could also serve as a free fatty acid binding site.

## Results

**Pharmacology of compound 1**. Identification and optimization of our initial A2 lead series focused on a careful examination of the three key areas of the pharmacophore consisting of the acidic head group, the center linker and the hydrophobic tail. Hypothesizing functional and structural similarities between endogenous FFAs and the literature GPR40 agonists led to the discovery of benzofurane acid derivatives as potent and highly selective GPR40 A2 agonists with a unique pharmacology (Supplementary Discussion). compound **1** does not bind the A1 site as TAK-875; but rather, it competes with AM-1638 for the A2 site (Fig. 1a). Interestingly, previous docking studies suggested that AM-1638 can bind to the same A1 site as TAK-875[14]; however, using the newly discovered A2 site in this report, we see that AM-1638 is better modeled in A2 when considering the internal ligand strain of its bound conformation in A1 (Supplementary Discussion). compound **1** is a full agonist (relative to the native ligand) that acts to increase intracellular $Ca^{2+}$ and cAMP levels as a result of GPR40 coupling to both $G\alpha_q$ and $G\alpha_s$ (Fig. 1b, c). In intraperitoneal glucose tolerance test (IPGTT), compound **1** dose-dependently increases insulin plasma levels and improves glucose metabolism (Fig. 2a–d). The absence of increased insulin levels by compound **1** in GPR40 knock-out (KO) mice indicates that the ability of compound 1 to increase insulin plasma level is GPR40-mediated (Fig. 2c). compound **1** also exhibits similar dose-dependent reduction of blood glucose in oral glucose tolerance test (OGTT) that is GPR40-mediated (Fig. 2e–g). In

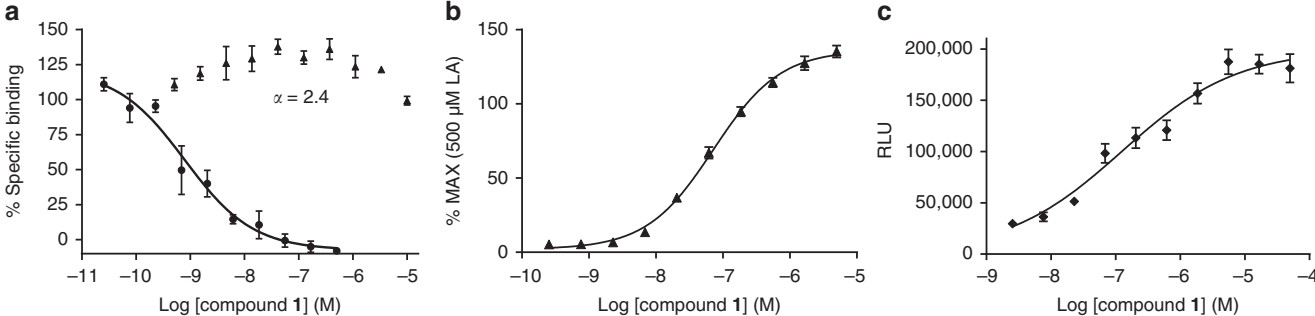

**Fig. 1** Ligand binding and functional assays of compound **1**. **a** Competitive radioligand binding for compound **1**. GPR40 membranes (prepared from stably transfected cells) were incubated with either [³H]-TAK-875 (triangles), $n = 3$, or [³H]-AM-1638 (circles), $n = 4$, in the presence of unlabeled compound **1**, as described in the Methods section. Percent specific binding (y-axis) was plotted against the log concentration of compound **1** (x-axis) and fit with a four parameter nonlinear logistic curve function with variable slope (GraphPad Prism, v7.00). The $K_i$ for compound **1** in [³H]-AM-1638 binding was 0.81 nM. An inhibition curve for [³H]-TAK-875 binding was not drawn since the addition of compound **1** resulted in enhanced binding activity. Data points represent the mean of single concentration determinations of independent experiments performed on different days with the same stock sample of compound. Error bars indicate the s.e.m. The affinity modulation factor (α) was calculated using the allosteric modulator titration equation within GraphPad Prism. **b** $Ca^{2+}$ mobilization assay (FLIPR®) using stably transfected cells. GPR40- $G\alpha_q$ signaling by compound **1** (triangles), $n = 4$. compound **1** acts as a full agonist relative to 500 μM of the natural ligand, linoleic acid. (EC₅₀ ~72 nM; Top = 136.3%; error bars indicate s.e.m.) **c** cAMP accumulation assay using stably transfected cells. Level of cAMP is measured as relative luminescence units (RLU). GPR40-$G\alpha_s$ signaling by compound **1** (diamonds), $n = 6$. (EC₅₀ ~125.8 nM; error bars indicate s.e.m.)

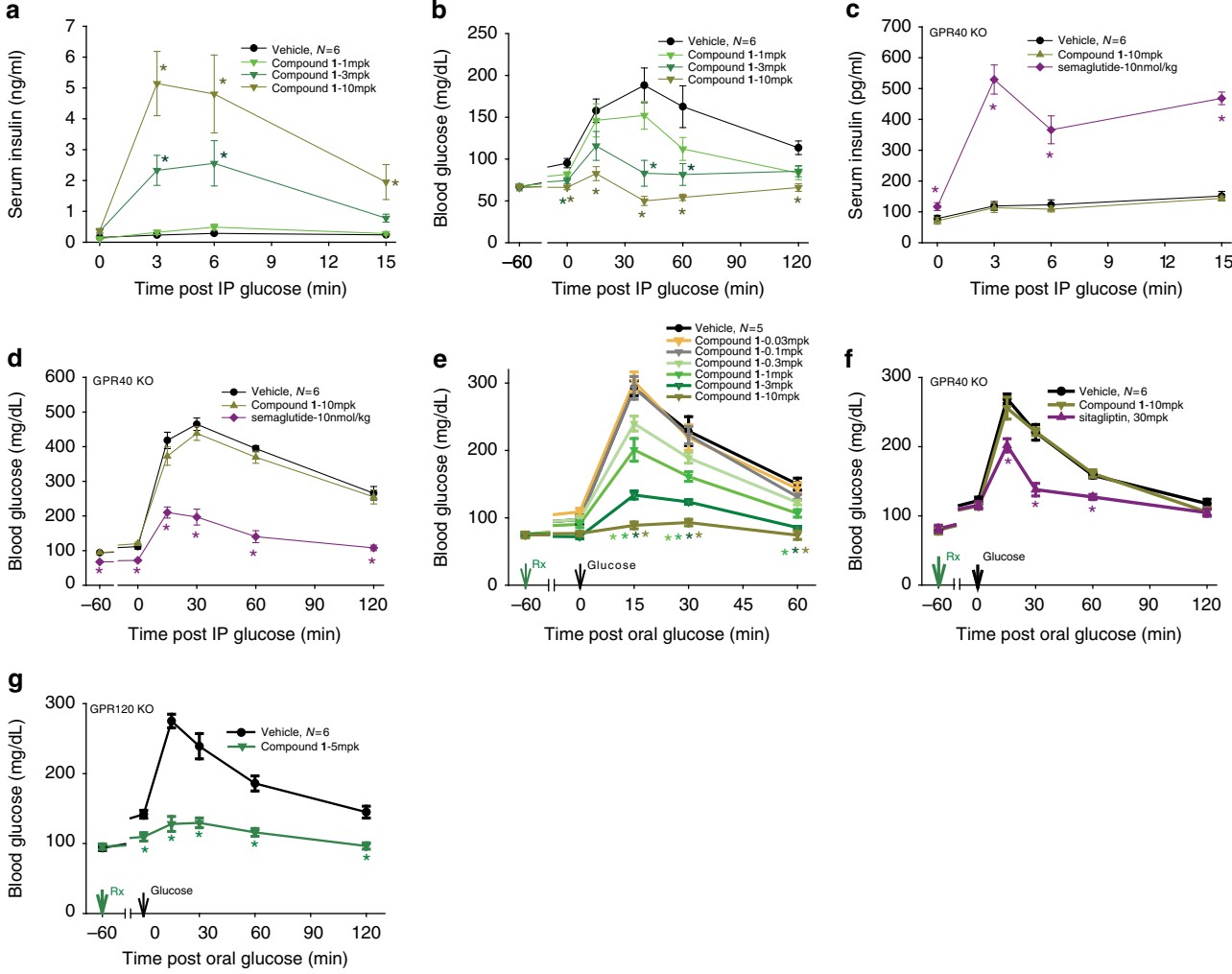

**Fig. 2** Glucose tolerance tests in mice. **a**, **b** Intraperitoneal glucose tolerance test (IPGTT) in wild type ICR mice. Dose-dependent increases in insulin secretion (**a**) and decreases in glucose excursion (**b**) are seen after treatment with various doses of compound **1**. **c**, **d** IPGTT in GPR40 knock-out (KO) mice. Insulin secretion (**c**) and glucose lowering (**d**) are not observed with high dose of compound **1**, but can be triggered with semaglutide treatment. **e** Oral glucose tolerance test (OGTT) in wild type C57BL/6 mice. Dose dependent decreases in glucose excursion are seen after treatment with various doses of compound **1**. **f** OGTT in GPR40 KO mice. The KO mice respond to glucose lowering by sitagliptin. No change in blood glucose is seen after treatment with high dose of compound **1**. **g** OGTT in GPR120 KO mice. A significant decrease in glucose excursion is seen after treatment with compound **1** at 5 mg/kg. $*p < 0.05$ vs vehicle by ANOVA with repeated measures followed by Dunnett's comparison

addition, oral administration of compound **1** demonstrates robust increase in incretins (GLP-1 and GIP) that is GPR40-mediated (Fig. 3a–d). compound **1** does not stimulate peptide YY (PYY) release in vivo (Fig. 3e) suggesting that increased incretin secretion is mechanism mediated instead of degranulation. In contrast, TAK-875 does not compete with AM-1638 for the A2 site; it only activates the $G\alpha_q/Ca^{2+}$ pathway and has very little incretin stimulation activity[14] (Supplementary Fig. 1a–d).

**Crystal structure of the GPR40-compound 1 complex.** The structure described here was obtained using the same stabilized human GPR40 construct as the GPR40-TAK-875 structure[17] (using the Protein Data Bank accession number '4PHU' in later discussion). This GPR40 receptor has a T4 lysozyme (T4L) protein inserted into the third intracellular loop (ICL3), as well as four point mutations that increase expression and thermal stability. These modifications likely constrain GPR40 in an inactive conformation, which could explain the reduced relative binding of [³H]-AM-1638 to the stabilized receptor vs. wild type (Supplementary Fig. 3b). The purified GPR40-compound **1** complex

was crystallized in lipidic cubic phase. Only a single crystal was needed to obtain a complete X-ray data set to 2.76 Å using synchrotron radiation, with apparently minimal or no radiation damage as judged by minimal change in the relative B factors of diffraction frames over data collection time[18] (Table 1; Supplementary Fig. 7).

The structure shows that compound **1** is highly lipid-exposed and does not participate in any packing interaction in the crystal (Supplementary Fig. 2a). This extra-helical allosteric binding site (A2) is a lipid-facing elongated hydrophobic pocket defined by transmembrane helices 3–5 (H3, H4, H5) and ICL2 (Figs. 4b and 5a). Some other examples of ligand binding to a location entirely outside of the helical bundle are provided by the P2Y₁ receptor (P2Y₁R) in complex with BPTU[19] and the glucagon receptor in complex with MK-0893[20]. In contrast to the effect of compound **1** as a GPR40 full agonist, both BPTU and MK-0893 are allosteric antagonists for their respective receptors and do not share the same binding position as compound **1** (Fig. 5c).

The A2 site occupied by compound **1** is distinct from the site occupied by TAK-875, and in 4PHU, the A2 site is occupied by a

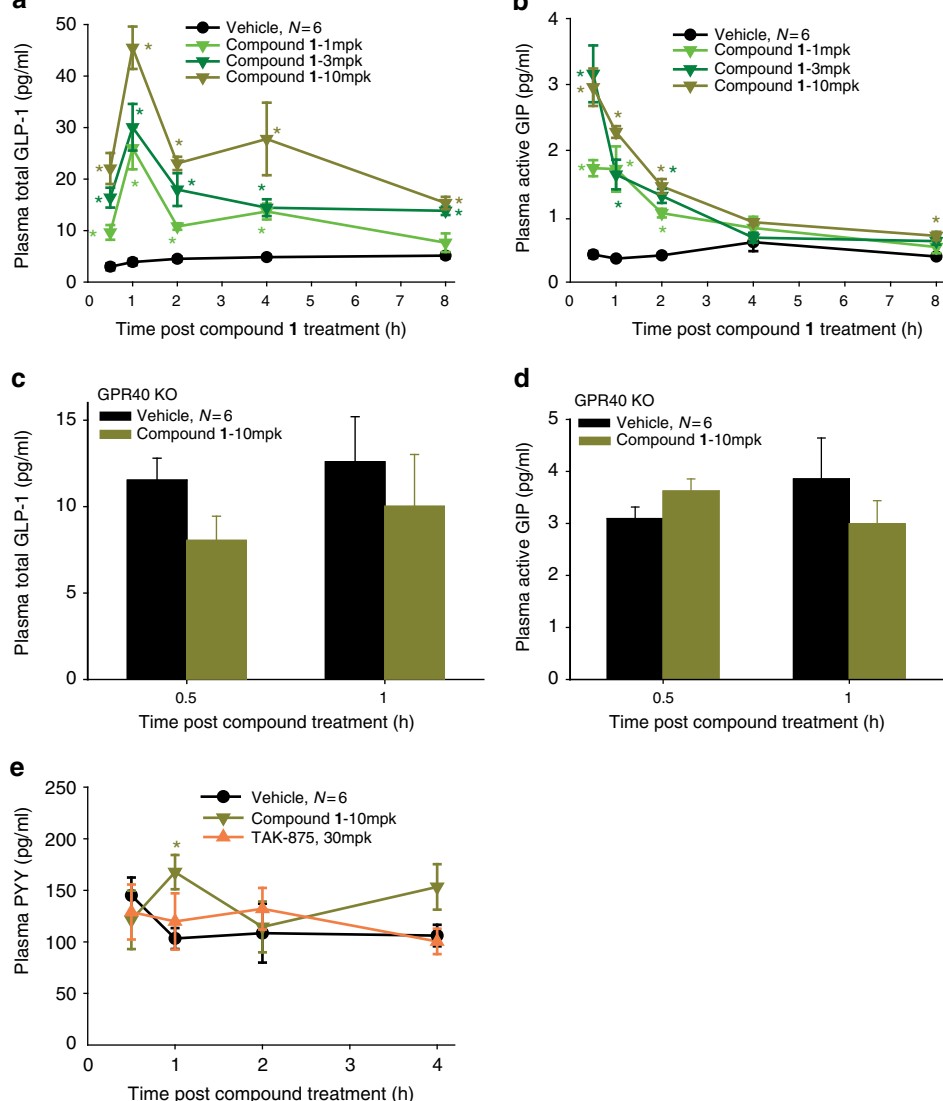

**Fig. 3** GLP-1 and GIP secretion by compound **1**. **a** compound **1** significantly increases plasma total GLP-1 concentrations up to 8 h. **b** compound **1** significantly increases plasma active GIP concentrations up to 8 h. **c**, **d** compound **1** has no effect on plasma total GLP-1 (**c**) and active GIP (**d**) concentrations in GPR40 KO mice. **e** Both compound **1** and TAK-875 exert no effect on plasma PYY level. Incretin values were log2 transformed and analyzed using ANOVA. *$p < 0.05$ vs vehicle by ANOVA with repeated measures followed by Dunnett's comparison

monoolein molecule, but the omit map electron density for monoolein is weak and discontinuous (Fig. 4a, b). The A2 site is located in the non-polar region of the receptor, so it was unexpected to find compound **1**, a ligand with a carboxylic acid moiety, binding at this location given the penalty of desolvation upon binding. However, the acidic moiety of compound **1** is fully satisfied by four hydrogen bond interactions: three from the side chains of Tyr 44$^{2.42}$, Tyr 114$^{ICL2}$, Ser 123$^{4.42}$ and one from water on the cytoplasmic side of the receptor (Fig. 5b) (superscripts indicate Ballesteros–Weinstein numbers[21]). Single-mutant study of Y44F, Y114F, and S123A in GPR40 shows that a simple change of removing the polar hydroxyl group in each of the three residues resulted in significant mitigation of the downstream Gα$_s$ stimulation by compound **1** (Fig. 6c) as a result of reduced compound **1** binding at the A2 site (Supplementary Fig. 3b, c). This demonstrates the importance of these three polar residues in engaging a ligand with acidic moiety in this hydrophobic environment. In addition, the section of compound **1** starting from the benzofuran moiety to the distal anisole

(methoxybenzene) provides additional affinity by maintaining good van der Waals (vdW) contact against H3 (Fig. 5a).

**Structural comparison with the GPR40-TAK-875 complex.** Comparing the A1 site occupied by TAK-875 in 4PHU, we observe that the entrance into the A1 site of our structure cannot accommodate TAK-875 due to movement of the upper half of H3 and H4 (Supplementary Fig. 6a–c), which is affected by crystal packing (Supplementary Fig. 2b); therefore, we believe that this slightly collapsed A1 site in our structure should not be inter-preted as a conformation change resulting from the binding of compound **1** in the A2 site. In 4PHU, Arg 183$^{5.39}$ and Arg 258$^{7.35}$ are the key basic residues that bind the acidic moiety of TAK-875; in our structure, both arginines hydrogen bond to the residues on extracellular loop 2 (ECL2), thereby pulling the ECL2 towards H6 and H7 (Supplementary Fig. 6d, e).

Another significant difference between 4PHU and our structure lies in the conformation of ICL2. In our structure, ICL2 adopts a short helix, but it is disordered in 4PHU (Fig. 4a, b). This difference in ICL2 structure is likely due to the hydrogen

**Table 1 Data collection and refinement statistics**

|  | GPR40-compound 1 |
|---|---|
| Data collection |  |
| Space group | C2 |
| Cell dimensions |  |
| $a, b, c$ (Å) | 147.7, 56.0, 80.3 |
| $\alpha, \beta, \gamma$ (°) | 90, 93.8, 90 |
| Resolution (Å) | 19.81–2.76 (2.91–2.76)* |
| $R_{sym}$ or $R_{merge}$ | 0.13 (0.64)* |
| $I/\sigma I$ | 5.9 (1.2)* |
| Completeness (%) | 99.6 (100)* |
| Redundancy | 3.7 (3.7)* |
| Refinement |  |
| No. reflections | 16,090 |
| $R_{work}/R_{free}$ | 0.23/0.27 |
| No. atoms |  |
| Protein | 3139 |
| Ligand/ion | 46 |
| Water | 52 |
| B-factors |  |
| Protein | 41.6 |
| Ligand/ion | 34.4 |
| Water | 34.2 |
| R.m.s deviations |  |
| Bond lengths (Å) | 0.006 |
| Bond angles (°) | 0.95 |

Standard definitions were used for all parameters
*Highest resolution shell is shown in parenthesis

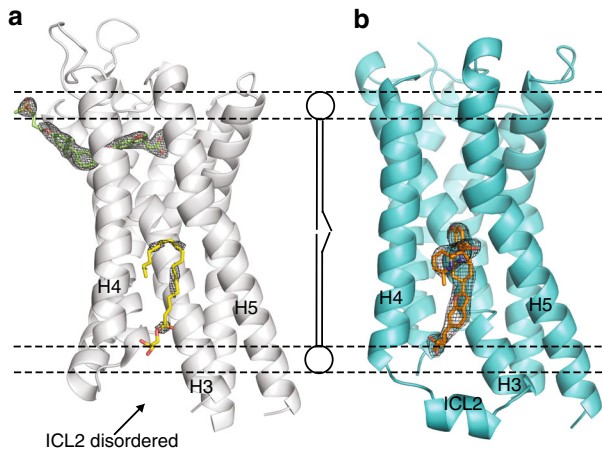

**Fig. 4** X-ray crystal structures of human GPR40-TAK-875 (PDB: 4PHU) and GPR40-compound **1** complexes. **a** GPR40 (gray ribbon) in complex with TAK-875 (green stick) in site A1. Monoolein is modeled in site A2 (yellow stick) in 4PHU. **b**, GPR40 (cyan ribbon) in complex with compound **1** (orange stick) in site A2. Transmembrane helices H3, H4, H5 and intracellular loop 2 (ICL2) are labeled. Phospholipids are rendered as balls and lines, illustrating the transmembrane region of the receptor. The $F_o$-$F_c$ omit maps of TAK-875, monoolein, and compound **1** were calculated in their absence and are shown as black mesh and contoured at 3.0 $\sigma$. Figures were prepared using PyMOL (Schrödinger, New York). Stereo image of **b** is provided in Supplementary Fig. 8

bond formed between the carboxylate moiety of compound **1** and Tyr 114[ICL2] (Fig. 5b). In the absence of this polar interaction, this helical conformation of ICL2 is unfavorable due to the fact that the polar side chain of Tyr 114[ICL2] cannot form a hydrogen bond in this hydrophobic region of site A2.

**The interaction between ICL2 of GPR40 and Gα$_s$.** The stabilization of ICL2 into a short helix by compound **1** (and presumably other A2 binders) could explain the enhanced G-protein coupling to the Gα$_s$/cAMP pathway by these compounds. In the X-ray structure of the active β2 adrenergic receptor (β2AR)●G protein complex[22], Phe 139[ICL2] of β2AR forms an important contact with a hydrophobic pocket from the Gα$_s$, and Leu 112[ICL2] in GPR40 is located in approximately the same position as the Phe 139[ICL2] of the active β2AR (Fig. 6a, b). Previous mutagenesis work in β2AR showed that a bulky hydrophobic amino acid at this position is needed for effective G protein coupling[23]. Similar "gain of function" of Gα$_s$ activity was also reported in the prostaglandin EP3 receptor by a single mutation in its ICL2[24]. To determine if a bulky hydrophobic residue in this area is necessary for GPR40 to couple to Gα$_s$, we performed cAMP accumulation assays in GPR40 with the L112A[ICL2] and the L112F[ICL2] mutants. Both mutants were assessed to still have near wild type A2 binding (Supplementary Fig. 3b), but no observable Gα$_s$ stimulation was detected in the L112A[ICL2] mutant with compound **1** up to micromolar concentrations; and in the the L112F[ICL2] mutant, higher E$_{max}$ and 3-fold improvement in EC$_{50}$ with compound **1** were observed over wild type (Fig. 6c). Interestingly, binding of TAK-875 in the L112F[ICL2] mutant also showed some Gα$_s$ stimulation, but only at high micromolar concentrations (Fig. 6d; Supplementary Fig. 3a). This suggests that the downstream Gα$_s$ activity by GPR40 can be modulated by altering the hydrophobic interface between ICL2 of GPR40 and Gα$_s$. Such modulations of G protein coupling using ICL2 chimeras, as well as site-directed mutagenesis in several other GPCRs have also been reported

(briefly reviewed by Zheng et al.[25]). Therefore, We hypothesize that full agonist binding at the A2 site may impart the Gα$_s$ in addition to the Gα$_q$ signaling due to the stabilization of the ICL2 of GPR40 in the conformation favorable for Gα$_s$ coupling. However, since our structure is in the inactive conformation, other unobserved conformational changes directed by A2 binders may also influence the additional Gα$_s$ activity.

## Discussion

To determine if the A2 site in GPR40 is also present in other GPCRs, we examined sequence alignments of four lipid GPCR subgroups (FFAR1–4, sphingosine-1-phosphate receptor 1–5, lysophosphatidic acid receptor 1–6, and prostaglandin E receptor 1–4). While none of these lipid receptors possess the same three amino acids in the equivalent positions of Tyr 44[2.42], Tyr 114[ICL2], and Ser 123[4.42] in GPR40, there are other polar residues present in this region that could still interact with a charged ligand (Supplementary Fig. 4a–c). Indeed, even though P2Y$_1$R is not a lipid GPCR, in the X-ray crystal structure of P2Y$_1$R complexed with BPTU[19] (Protein Data Bank accession number 4XNV), a cholesteryl hemisuccinate (CHS) molecule was modeled in this location with relatively weak omit map electron density (Supplementary Fig. 5c). In this P2Y$_1$R structure, the hydrophobic portion of the CHS maintains some vdW contact in the pocket and its acidic moiety is partially satisfied by hydrogen bonds with the side chains of Tyr 89[2.42] and His 148[3.49] (Supplementary Fig. 5a, b). It is less certain if there is any polar interaction from the receptor to the other oxygen atom of the ligand's acidic moiety due to the fact that part of the ICL2 is disordered in the structure, and Ser 151[3.52] (another nearby polar amino acid) is too far away from forming a productive hydrogen bond with the acidic moiety (~4 Å) (Supplementary Fig. 5b). Notably still, Tyr 89[2.42] of P2Y$_1$R corresponds to Tyr 44[2.42] in GPR40, and His 148[3.49] of P2Y$_1$R corresponds to Gly 103[3.49] in GPR40 or to Asp in the DRY-motif in most GPCRs (reviewed in ref. [26]). This demonstrates that the relatively sparse

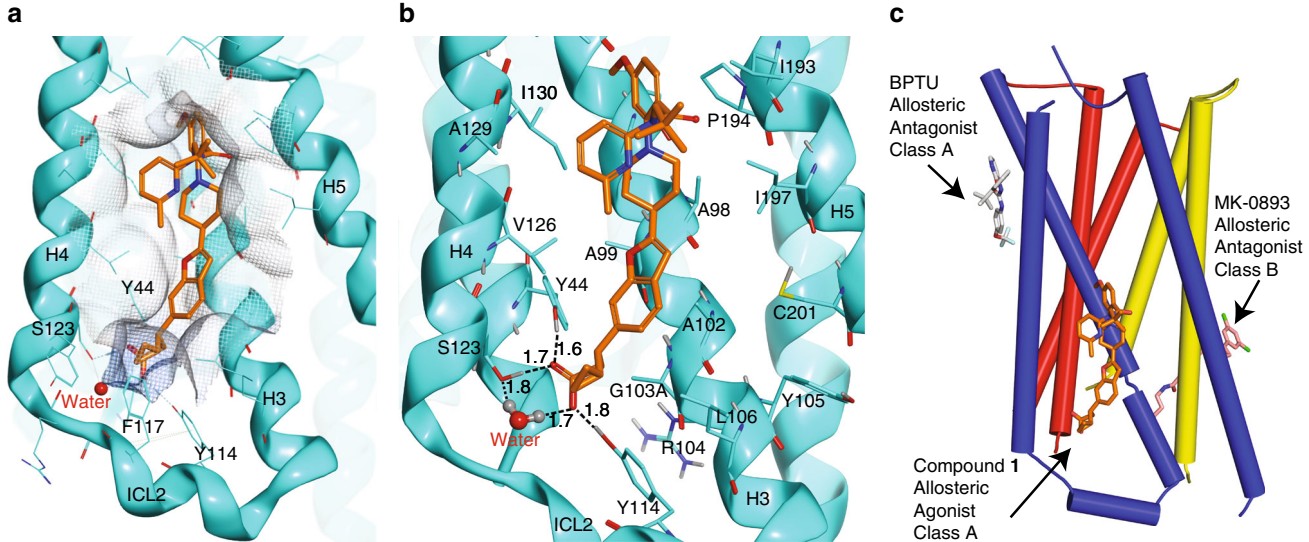

**Fig. 5** Binding interaction of compound **1**. **a** In silver mesh is an "interaction surface" of site A2, indicating that the ligand is within optimal van der Waals (vdW) contact distance when touching the surface (like the seven-position of the benzofuran). Receptor is colored in cyan, and compound **1** in orange. Ribbon represents the transmembrane helices. **b** A more detailed view of the binding site. Hydrogen bonds are represented as dotted lines. Distances shown are between oxygen and hydrogen. Hydrogens were computationally modeled into the structure to illustrate hydrogen bond donors and acceptors. Receptor side chains are colored by atom type (carbon in cyan, oxygen in red, nitrogen in blue, hydrogen in gray, and sulfur in yellow). compound **1** is illustrated as sticks, colored by atom type with carbon in orange. Transmembrane helices H3, H4, H5 and ICL2 are labeled. Note that G103A is one of the stabilizing mutations. A103 is nearby, and its methyl group sidechain forms a vdW contact with the carbon alpha to the acidic group in compound **1**. **c** Schematic overview of identified GPCR ligand binding positions outside the helical bundles. Transmembrane helices H1 and H2 are colored in red; H3, H4, and H5 are colored in purple; H6 and H7 are colored in yellow. Figures were prepared using Molecular Operating Environment (MOE)[35] and PyMOL (Schrödinger, New York)

presence of polar amino acids in this non-polar region of the receptor may still provide specific polar interaction with a charged ligand. Further study is required to assess if such allosteric binding site exists for P2Y$_1$R.

The total number of carbon atoms from the carboxylic group to the o-methyl of compound **1** and AM-1638 can be counted as 17 atoms, similar to the length of the 18-carbon FFA native ligand: γ-linolenic acid (γ-LA) (Fig. 7a–c). Supported by the observation of a lipid molecule (monoolein) in the same location in 4PHU (Fig. 4a), we hypothesized that the A2 site could also be a second lipid binding site. To assess the likelihood of an 18-carbon FFA being able to bind at site A2, we modeled both stearic acid (C18) and γ-LA (C18:3) into the site and estimated the free energy for each ligand going from the unbound to the bound state. While both ligands fit the A2 site, the bound state of γ-LA appeared to show more favorable enthalpy of binding, with two olefins forming hydrogen bonds with backbone oxygens of the protein in a very non-polar environment. In addition, γ-LA, as expected due to the partial unsaturation, appeared to access far fewer conformations in the unbound state than stearic acid, resulting in a much smaller entropy loss to adopt the bound state conformation. Binding of γ-LA, which has been reported[6] to have a better GPR40 affinity than stearic acid, appears to originate from better enthalpy and entropy due to the partial unsaturation. γ-LA is a full agonist of GPR40 and has positive cooperativity with TAK-875 in the Ca$^{2+}$ mobilization and insulin secretion assays[13]. It is plausible that the allosteric activity observed between TAK-875 and γ-LA is mediated by cooperativity between sites A1 and A2 as γ-LA can be modeled in A2 (Fig. 7d). Moreover, we also observed positive allosteric cooperativity between TAK-875 and compound **1** ($\alpha = 2.1$; $\beta = 14$) as well as enhanced thermal stability when both ligands are present (Fig. 8a, b). After several rounds of crystallizations trials of GPR40 complexed with both TAK-875 and compound **1**, we were not able to

obtain crystals. A reasonable model of this two-ligand ternary complex can be presented where there is no overlap of the two allosteric binding sites (Fig. 8c). Indeed, more studies remain needed to extensively characterize the allosteric relationship of the A1 and A2 sites and how their allostery modulates the downstream Gα$_q$ and Gα$_s$ signaling pathways. It is also worth speculating that GPR40 may not be a receptor that has a dedicated orthosteric site as previously hypothesized[10,16] but a receptor with two allosteric binding sites that can both be accessed by endogenous FFAs.

Our findings, as well as other recently reported agonist-bound GPR40 structures[27] provide structural evidence that agonists binding distinct allosteric sites of the same receptor generate biased G-protein signaling. This allosteric mechanism expands our understanding of basic pharmacology and could provide insight into the function of other members of the GPCR family. In addition, these results may be useful for the rational design of better GPR40 agonists. The possibility that oral administration of a GPR40 agonist can lead to increases in circulating levels of the therapeutic biomolecules insulin and GLP-1 has the potential to improve future therapies for T2DM.

## Methods

**Ligand-binding assay of the wild type and mutant receptors**. Crude cell surface membranes were prepared from human embryonic kidney 293 (HEK 293) cells stably transfected (for the wild type) or transiently transfected (for the mutants and corresponding wild type) with full length recombinant human GPR40 (GPR40) cDNA, using differential centrifugation methods.

A1 Site Binding. 10 μL of compound diluted in 100% DMSO and 90 μL of Assay Buffer (50 mM Tris-HCl, pH 7.5, 5 mM CaCl$_2$, 5 mM MgCl$_2$, 0.1% w/v fatty acid-free BSA) were added to a deep 96-well polypropylene assay plate (Beckman Coulter). In total 200 μL of [$^3$H]-TAK-875 (52 Ci/mmol, Quotient Bioresearch Radiochemicals, Ltd.; 5 nM final concentration) and 200 μL of GPR40 membranes (5 μg/well), both diluted in Assay Buffer, were added to the assay plate, followed by a 1 min shake and a 2 h incubation at room temperature (22 °C).

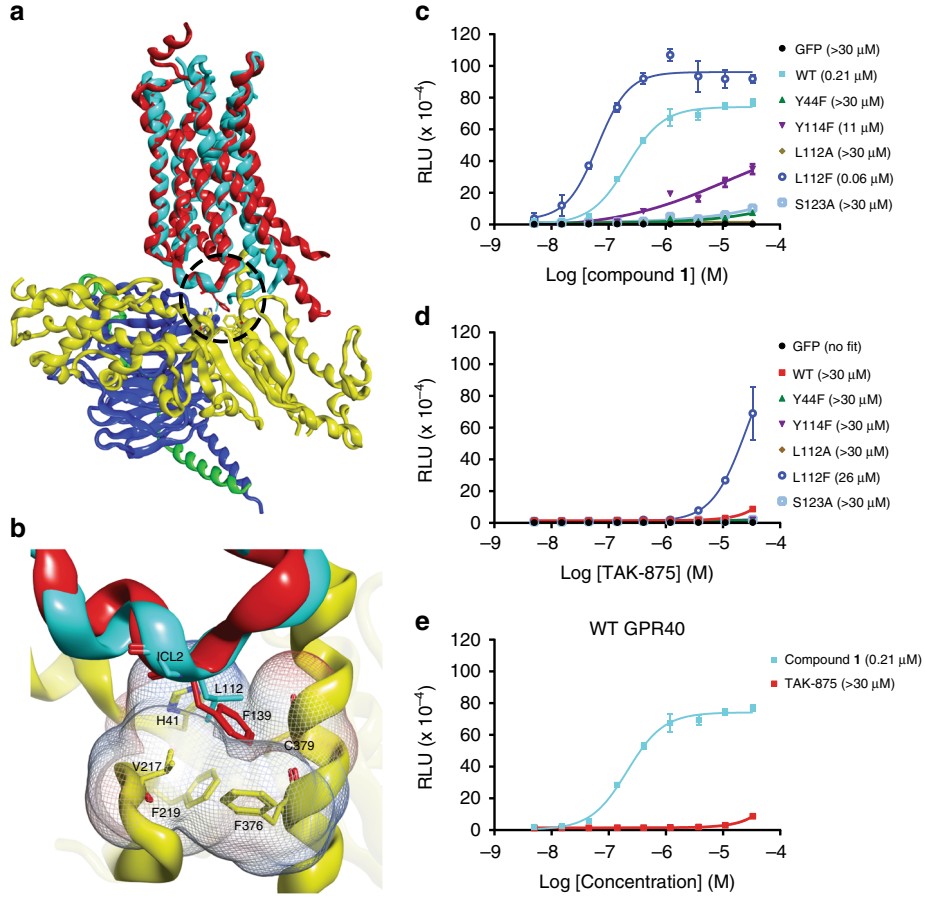

**Fig. 6** Functional study on GPR40 site A2 mutants and intracellular loop 2 (ICL2) mutants. **a** Structural superposition of GPR40-compound **1** with β2AR•G protein complex (PDB: 3SN6) in ribbon representation (Cα RMSD = 1.31 Å). GPR40-compound **1** structure is in cyan; compound **1** is shown as orange stick; β2AR is in red; Gα$_s$ is in yellow; Gβ is in purple; Gγ is in green. The vdW interaction surface of the A2 binding site is shown as mesh in silver. The region circled is magnified in **b**. **b** An important hydrophobic contact between the ICL2 of β2AR and Gα$_s$. The ICL2 is labeled. vdW interaction surface formed by H41, V217, F219, F376, and C379 of the Gα$_s$ is shown as mesh and colored according the electrostatic charge of the residues (red: negative; blue: positive; white: neutral). **c, d** cAMP accumulation assays in the presence of compound **1** (**c**) or TAK-875 (**d**). Levels of cAMP are measured as relative luminescence units (RLU). Y114F, S123A, Y44F are site A2 mutations. L112A and L112F are ICL2 mutations involved in Gα$_s$ coupling. $n = 3$. EC$_{50}$ values are shown in the parentheses. Error bars indicate s.e.m. **e** Measurements in WT extracted from **c** and **d** for ease of comparison

A2 Site Binding. 10 μL of compound diluted in 100% DMSO and 90 μL of Assay Buffer (50 mM Tris-HCl, pH 7.5, 10 mM CaCl$_2$, 10 mM MgCl$_2$, 0.1% w/v Bacitracin, 2.5% w/v dextran charcoal-treated FBS) were added to a deep 96-well polypropylene assay plate (Beckman Coulter). Two hundred μl of [$^3$H]-AM-1638 (84 Ci/mmol, Quotient Bioresearch Radiochemicals, Ltd.; 0.5 nM final concentration) and 200 μL of GPR40 membranes (5 μg/well), both diluted in Assay Buffer, were added to the assay plate, followed by a 1 min shake and a 1 h incubation at room temperature (22 °C).

Assays were terminated by filtration through GF/C glass fiber filtermats (Perkin Elmer) presoaked in 50 mM Tris–HCl, pH 7.5, using a Mach III cell harvester (TomTec). Filtermats were washed two times with 5 mL of ice-cold 50 mM Tris–HCl, pH 7.5 buffer, dried 1 h in a convection oven at 60 °C and embedded with Meltilex A solid scintillant (Perkin Elmer). Radioactivity was determined as counts per min (cpm) using a Trilux Microbeta plate scintillation counter (Perkin Elmer). The equilibrium dissociation constant ($K_i$) was calculated from the relative IC$_{50}$ value based upon the equation $K_i = IC_{50}/(1+L/K_d)$, where L equals the concentration of radioligand used in the experiment and $K_d$ equals the equilibrium binding affinity constant of the radioligand, determined from saturation analysis ([$^3$H]-TAK-875 $K_d$=6.2 nM; [$^3$H]-AM-1638 $K_d$=1.2 nM).

**Ca$^{2+}$ mobilization assay (FLIPR®)**. HEK293 cells stably overexpressing full length human GPR40 were plated (50K cells/well) into 96-well microtiter plates using Dulbecco's Modified Eagle's Medium (DMEM) supplemented with 10% certified fetal bovine serum (FBS), 20 mM HEPES, pH 7.4, 1 mM Sodium Pyruvate, 100 U/mL Penicillin, 100 μg/mL Streptomycin, and 800 μg/mL Geneticin. The cells were incubated overnight at 37 °C and 5% CO$_2$. Calcium 4 dye (Molecular Devices) diluted in assay buffer was added (100 μL per well) to the cell plates, followed by a 1 h incubation in the dark at 25 °C. Test compounds were serially diluted three-fold in 100% DMSO and immediately diluted in assay buffer. Diluted compounds were

immediately added to the cell plates using the liquid handling capabilities of a FLIPR to achieve final top test compound concentrations of 10 μM for TAK-875 or 5 μM for compound **1** (10-point concentration response curve) at a final DMSO concentration of 1%. Receptor activation was immediately measured as an increase in intracellular calcium using the FLIPR® over 3 min. To determine agonist responses, relative fluorescence units (RFUs) over 60 reads were calculated per well and used to calculate percent stimulation relative to 500 μM of the natural ligand, linoleic acid, response. EC$_{50}$ values were calculated by plotting test compound concentration vs. percent stimulation using a 4-parameter logistic curve fitting equation.

**cAMP accumulation assay**. HEK293 cells transiently transfected with Promega's cAMP response element (CRE/luc2P) and a GPR40 construct were plated (40K cells/well) into 96-well microtiter plates using Dulbecco's Modified Eagle's Medium (DMEM) supplemented with 10% FBS, 20 mM HEPES, pH 7.4, 100 U/mL Penicillin, 100 μg/mL Streptomycin, and 800 μg/mL Geneticin. Test compounds were serially diluted three-fold in 100% DMSO and immediately diluted in media. Diluted compounds were added to the cell plates to achieve final top test compound concentration of 100 μM for compound **1** (10-point concentration response curve) and incubated for 4 h. cAMP levels were indirectly measured using the Promega's Bright-Glo Luciferase Assay System. EC$_{50}$ values were calculated using a four-parameter logistic curve fitting equation.

**Statistics and reagents of in vitro assays**. All in vitro assays were developed according to the procedures outlined in the Assay Guidance Manual[28]. Technical replicates were performed on separate days and the number of replicates required for statistical significance was based on Minimum Significant Ratios[29] established during assay development. Extreme outliers were excluded when they were distant from other observations in the same data set and no pre-established criteria existed.

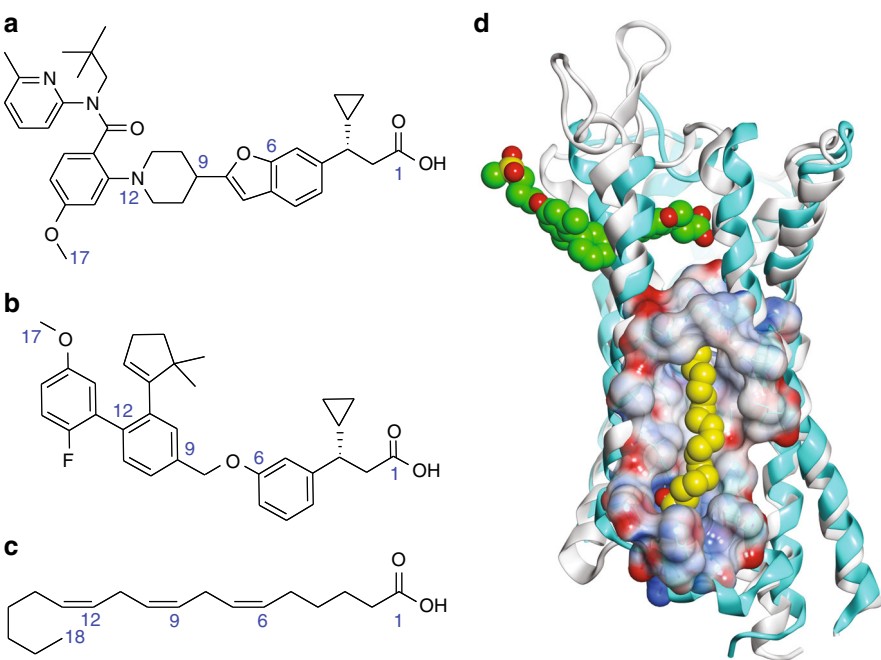

**Fig. 7** Site A2 as a possible free fatty acid binding site. Chemical structure of: **a** compound **1**; **b** AM-1638; **c** γ-linolenic acid (γ-LA). Some of the atoms are numbered to show length similarity between compound **1**, AM-1638, and γ-LA. **d** Structural superposition of 4PHU with the GPR40-compound **1** complex (Cα RMSD = 0.99 Å) in ribbon representation. 4PHU is in white with TAK-875 shown in green as space-filling representation. GPR40-compound **1** structure is in cyan, and compound **1** in site A2 is replaced by a model of γ-LA in yellow as space-filling representation. The A2 binding pocket is rendered as Connolly analytic surface[44], colored based on electrostatic potential (red: negative; blue: positive; white: neutral)

Parental HEK293 cell lines (obtained from ATCC) were used because of the recombinant DNA transfection ease and robust growing conditions. HEK293 cell lines were not authenticated independently (e.g., using Short Tandem Repeat DNA profiling), but they were not used as a model for kidney cells or kidney function. Cell lines were tested and found to be mycoplasma negative (MycoAlert®, Lonza Rockland, Inc).

**Animals**. Eight to nine week old male C57BL/6 mice and nine to ten week old male ICR mice were obtained from Envigo (Indianapolis, IN). Nine to thirteen weeks old GPR40 and GPR120 knock-out (KO) mice were generated by Taconic (Hudson, NY). All animals were singly housed in a temperature-controlled (24 °C) facility using a 12-h light/dark cycle (light on 0600) and had ad libitum access to food and water unless otherwise stated. All in vivo experiments were performed in compliance with the policies of the Animal Care and Use Committee of Eli Lilly and Company, in conjunction with the American Association for the Accreditation of Laboratory Animal Care-approved guidelines.

**Intraperitoneal glucose tolerance test (IPGTT) in lean mice**. Nine to ten week old male ICR mice (Envigo) and GPR40 KO mice (Taconic) were used. The day before an IPGTT (~1600), animals were fasted in clean cages. The following morning (~0800), animals were randomized on fasting glucose and body weight ($N = 6$ per group). Mice were orally administered a test article or vehicle 60 min prior to an intraperitoneal (IP) injection of glucose (2g/kg). Blood glucose levels were determined by a glucometer from tail bleeds taken at 0, 15, 30, 60, and 120 min after the glucose challenge. An average of two readings were reported at each time point. Also, blood samples were collected into serum tubes at 0, 3, 6, and 15 min post glucose injection for insulin measurement. Serum tubes were centrifuged at 3000*g* for 5 min and serum transferred into 96-well plates for insulin analysis by a Mesoscale rat/mouse insulin assay.

**Oral glucose tolerance test (OGTT)**. Eight to thirteen week old male C57BL/6 or KO mice were used. The day before the study, at 1600 hours, animals were transferred into clean cages and food was removed. The following morning (~0800 hours), mice were randomized by block randomization to treatment groups ($N = 5$–7 per group) so each group had similar mean and standard deviation of body weight and blood glucose. Sample size calculation suggests $N = 5$ per group is sufficient to achieve a power of 80% to detect 30% change in blood glucose. Animals were dosed with vehicle (0.5% MC/0.25% Tween-80) or the test article by oral gavage. Group allocations were not blinded from the experimenters who collected the data. After 60 min compound treatment animals were then given an oral gavage of glucose (2 g/kg). At 0, 15, 30, 60, and 120 min post oral glucose, two glucose readings via glucometers were obtained. The average of two glucose

readings was reported at each time point. Data were expressed as mean±standard error (s.e.m.). Blood glucose values were analyzed using ANOVA with repeated measures. Cook's distance was also calculated from linear regression. All animals were included in the analysis with their individual averaged Cook's distance < 1. Dunnett's comparison was performed at each time point by R program. Significance is denoted at $p < 0.05$.

**Incretin secretion assays**. Nine week old male C57Bl/6 mice (Envigo, Indianapolis) were used. The night before the incretin assay, mice were transferred to clean cages and fasted overnight. On the morning of the assay, the mice were weighed and randomized by block randomization into groups based on body weight ($N = 6$) so each group had similar mean and standard deviation of body weight. Sample size calculation suggests $N = 6$ per group is sufficient to achieve a power of 80% to detect 50% change in the GLP-1 level. Animals were dosed with vehicle or the test article by oral gavage. Group allocations were not blinded from the experimenters who collected the data. The mice were sacrificed at 0.5, 1, 2, 4, or 8 h post the compound treatment. Blood was taken by cardiac puncture into EDTA tubes containing DPPIV inhibitor and aprotinin cocktail after $CO_2$ euthanization. The final DPPIV inhibitor concentration was 50 μM and aprotinin was 250 K IU/mL. Blood samples were centrifuged at 3000×*g* for 5 min and plasma transferred into 96 well plates. Plasma total GLP-1 and active GIP levels were measured by ELISA assays that were developed in-house. Data were expressed as mean±s.e.m. Incretin values were log2 transformed as suggested by Box-Cox transformation and analyzed using ANOVA. Cook's distance was also calculated from linear regression. All animals were included in the analysis with their individual Cook's distance < 1. Significance is denoted at $p < 0.05$.

**Computational modeling of AM-1638 and γ-linolenic acid (γ-LA)**. γ-LA was modeled in site A2 of GPR40 by minimization in the binding site with two manually imposed hydrogen bond distance constraints from the electron-rich olefin carbons at positions 6 and 12 of the hydrocarbon tail to backbone carbonyl oxygens of residues 99 and 95 on GPR40, respectively. AM-1638 was modeled in Site A1 by alignment onto the experimental bound state of TAK-875 (Protein Data Bank accession number 4PHU) and in Site A2 by alignment onto the bound-state of compound **1**, followed by fully unrestrained minimizations. In order to estimate ligand strain for the predicted bound state of γ-LA, compared to the free state, the FreeForm utility within the SZYBKI[30] application was used. All molecular mechanics minimizations were run using the AMBER10:EHT force field[31,32], with a reaction field model treatment of non-bonded electrostatics[33,34] using interior and exterior dielectric constants of 1 and 4, respectively, as implemented in the MOE 2014.09 software[35].

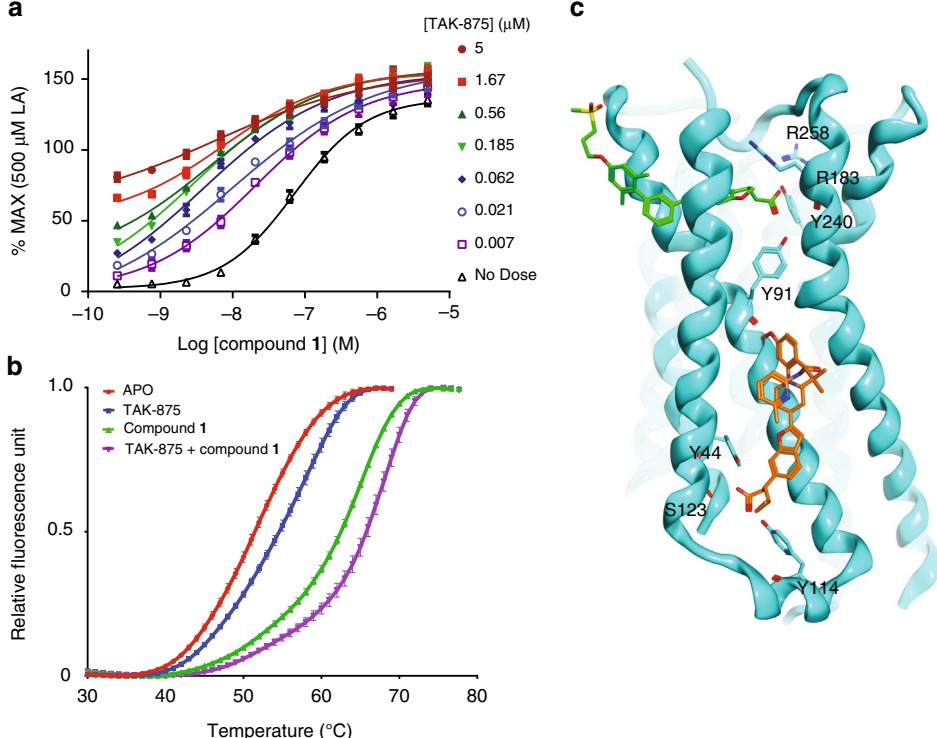

**Fig. 8** Positive cooperativity between TAK-875 and compound **1**. **a** $Ca^{2+}$ mobilization assay (FLIPR®) of compound **1** in the presence of various concentrations of TAK-875. The presence of TAK-875 shifts the dose response curve of compound **1** to the left in the FLIPR assay (from 72 to 5.8 nM), indicating induction of positive cooperativity. $N = 4$. Error bars indicate s.e.m. The cooperativity composite factor (Log $\alpha\beta = 1.46$) was calculated using the operational model for agonism as described by Wootten et al.[45], with the slope constrained to 1. Using an average $\alpha$-factor of 2.1 (averages from Figs. 1a and 8a), the efficacy modulation cooperativity factor ($\beta$) was calculated to be 14. **b** Thermal denaturation experiment of unliganded GPR40 ($T_m = 51.4$ °C) and liganded complexes ($T_m$ GPR40-TAK-875 = 58.4 °C; $T_m$ GPR40-compound **1** = 65.3 °C; $T_m$ GPR40-TAK-875-compound **1** = 68.2 °C). $N = 3$. Error bars indicate standard deviation. Higher $T_m$ values reflect higher thermal stability. **c** Structure of GPR40-compound **1** with TAK-875 modeled in site A1. Crystal packing artifact in GPR40-compound **1** looks to have stabilized a somewhat collapsed site A1 that appears incompatible with TAK-875 binding. To model this ligand's binding pose, we removed residues 72–86 from transmembrane helix 3 and extracellular loop 1 in GPR40-compound **1**, and modeled those residues back using the corresponding residues in 4PHU. The binding pose of TAK-875 from 4PHU was then superimposed in site A1. Hydrogen bond tethers were added from the carboxylate to the hydroxyl-hydrogen atoms of Tyr 91 and Tyr 240, followed by unrestrained minimization of the ligand, while also allowing the newly modeled region of GPR40 to move unrestrained. The resulting site A1 ends up larger than in our structure and smaller than that of 4PHU[17]

**Cloning and baculovirus expression**. Human GPR40 encompassing amino acids 1–211+L42A/F88A/G103A/Y202F and amino acids 214–300 with a T4 lysozyme internal fusion (GPR40 numbered relative to reference sequence NP_005294) was TOPO cloned using forward primer: ATGGATTACAAGGATGATGATGA and reverse primer: ATTAATGGTGATGGTGGTGATGATGGTG into a custom TOPO adapted pFastBac vector (Life Technologies). Further details on the mutation-stabilized GPR40-T4L crystallization construct can be found in the previous report[17]. Standard baculovirus expression using a modified version of the Bac-to-Bac system protocol (Life Technologies) in combination with the DH10EMBacY bacmid (Geneva Biotech) was used to generate virus. Fermentations of GPR40 in *Sf*9 cells at 2 million cells/mL were performed at 10 L or 24 L scale, on rocking platforms at 22 rpm, with a 12° angle, 27.5 °C and 0.5 min/L air, for a duration of 48 h, harvested by centrifugation, and pellets were stored at −80 °C for purification.

**Purification**. Frozen cell pellets were resuspended in a lysis buffer containing 20 mM HEPES, pH 7.5, 20 mM NaCl, 2 mg/mL iodoacetamide, protease inhibitor cocktail (Calbiochem), and Turbonuclease (Accelagen). Cells were lysed by dounce homogenization, and cell membranes were isolated by ultracentrifugation in a type 45 Ti rotor (Beckman Coulter) at ~224,000×*g* for 1.5 h. Membranes were washed by dounce homogenization in buffer containing 20 mM HEPES, pH 7.5, 500 mM NaCl, 10% glycerol, 2 mg/mL iodoacetamide, 20 mM imidazole, protease inhibitor cocktail, and turbonuclease, followed by another round of ultracentrifugation. After the high-salt wash, membranes were resuspended in the same buffer and stored at −80 °C.

Prior to solubilization, the membrane resuspension was incubated with 10 μM of compound **1** for 2 h to allow the receptor–ligand complex to form. A solution of 200 mM n-dodecyl-beta-D-maltoside (DDM; Anatrace) and 2% cholesterol hemisuccinate (CHS; Anatrace) was added to a final concentration of 10 mM DDM and 0.1% CHS, and mixed for 30 min at 4 °C with gentle stirring. The insoluble

material was removed by ultracentrifugation, and the supernatant was incubated with TALON resin (Clontech) for 2 h. Resin was packed in a gravity flow column, and detergent exchange was accomplished by directly washing the resin with 20 column volumes of 10 mM HEPES, pH 7.5, 10% glycerol, 300 mM NaCl, 5 mM imidazole, 0.1% lauryl maltose neopentyl glycol (LMNG; Anatrace), 0.01% CHS, protease inhibitor cocktail, and 10 μM compound **1**. Receptor was eluted with 10 mM HEPES, pH 7.5, 10% glycerol, 300 mM NaCl, 200 mM imidazole, 0.1% LMNG, 0.01% CHS, protease inhibitor cocktail, and 10 μM compound **1**. The buffer was exchanged by desalting column into 10 mM HEPES, pH 7.5, 300 mM NaCl, 0.01% LMNG, 0.001% CHS, protease inhibitor cocktail, and 10 μM compound **1**. PNGase and TEV protease were added to the desalted sample and incubated overnight at 4 °C to remove glycosylation and the C-terminal 8×His tag. Next morning, FLAG® resin (Sigma) was added to the mixture and incubated for 2 h; resin was washed with 10 mM HEPES, pH 7.5, 300 mM NaCl, 0.01% LMNG, 0.001% CHS, protease inhibitor cocktail, 10 μM compound **1** and then the receptor was eluted with 10 mM HEPES, pH 7.5, 300 mM NaCl, 0.01% LMNG, 0.001% CHS, protease inhibitor cocktail, 10 μM compound **1**, 100 μg/mL FLAG peptide. Purified receptor was concentrated in a 50 kDa MWCO filter (Millipore) to ~500 μL and further purified in a S200 10/300 (GE Healthcare) size exclusion column in 20 mM HEPES, pH 7.5, 300 mM NaCl, 0.01% LMNG, 0.001% CHS, 1 μM compound **1**. Receptors in the monodisperse fractions were pooled, and the ligand concentration was increased to 100 μM. The final sample was concentrated in a 50 kDa MWCO filter to 60–80 mg/mL for crystallization.

**Thermal stability assay**. 7-Diethylamino-3-(4-maleimidylphenyl)−4-methylcoumarin (CPM dye; Adipogene) thermal denaturation experiments[36] were carried out in 0.1 mL PCR strip tubes and measured with a Rotor-Gene Q real-time PCR instrument (Qiagen, Model 6-Plex). Assay was performed in a total volume of 20 μL using 5 μM purified unliganded GPR40 in 10 mM HEPES, pH 7.5, 150 mM NaCl, 1 mM DDM, 0.01% CHS, 50 μM ligand, 25 μM CPM dye. For the single

ligand assay, 50 µM ligand was used. In the double-ligand assay, 25 µM of compound **1** and 25 µM of TAK-875 were used. Samples were prepared as triplicates and incubated at 4 °C for 1.5 h in the dark before thermal denaturation. The excitation wavelength was set at 365 nm, and the emission wavelength was at 460 nm. Melts were performed over a temperature range of 25–90 °C, ramping 1 °C every 5 s. Melting curves were processed with GraphPad Prism program (GraphPadPrism v.6.04). The inflection point of the melting curves was used as the $T_m$ and was determined using the first derivative values in the Rotor-Gene Q real-time software (v. 2.3.1 (Build 49)).

**Crystallization**. compound **1** bound GPR40-T4L complex was mixed with monoolein containing 10% cholesterol in 1:1.5 parts v/v protein:lipid ratio using the twin-syringe mixing method[37]. Using a mosquito LCP crystallization robot (TTP labtech), 50 nL size LCP boluses were dispensed onto 96-well glass sandwich plates (Hampton) and overlaid with 0.8 µL of precipitant solution. Crystals reached full size in 10–12 days at 20 °C in the optimized condition containing 0.1 M Tris-HCl, pH 8.5, 30% PEG 400, 0.2 M ammonium formate. Crystals in LCP were harvested and flash frozen in liquid nitrogen without additional cryoprotectant.

**Data collection, processing and structure determination**. A complete X-ray data set was collected on a single rod-shaped crystal ($\sim 10 \times 70$ µm$^2$) with negligible radiation damage (Supplementary Fig. 7) at 100K in a single sweep of $180 \times 1°$ oscillations and 1.2 s exposure with an unattenuated beam at beamline LRL-CAT (31-ID-D) at the Advanced Photon Source in Argonne National Laboratory, Lemont, IL. The beam size was approximately $80 \times 70$ µm$^2$ FWHM (full width half maximum). The whole sample (i.e. the LCP sample/blob in the MiTeGen loop that contained the crystal) was exposed to the beam during data collection, and loop rastering was not performed. The diffraction data were integrated using autoP-ROC/XDS[38,39] and merged and scaled in SCALA[18] from the CCP4 suite[40].

Crystal structure of the GPR40-compound **1** complex was determined by molecular replacement using the previously solved human GPR40-TAK-875 structure: 4PHU[17]. GPR40 (intracellular and extracellular loops removed) and lysozyme were used as separate search models. Clear density for the ligand was observed immediately. After numerous cycles of refinement with REFMAC5[41] and model building with COOT[42], the models were refined to reasonable R factors. The structure was validated using MolProbity[43]. The Ramachandran plot reports 95.7% in the most favored region, 4.3% in the allowed region, and none in the disallowed region. For details, see Table 1.

**Synthesis of compound 1**. Described in detail in Supplementary Method section.

**Data availability**. Coordinates and structure factors of the GPR40-compound **1** X-ray structure have been deposited in the Protein Data Bank under accession number 5KW2. The PDB accession codes 4PHU and 3SN6 were used in this study. All relevant data are available from the corresponding authors upon request.

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

## Acknowledgements

We thank Han Wu for statistical assistance, and Justine Cudal, Erin Krause, Blake Gagner, Shelby Hedrick, and Derek Mori for technical support. We thank David B. Wainscott and Francis S. Willard for analyzing the data on ligand cooperativity and allostery. We thank Stephen Wasserman and the staff at the Lilly Research Laboratories Collaborative Access Team (LRL-CAT) beamline Sector 31 of the Advanced Photon Source. This research used resources of the Advanced Photon Source, a U.S. Department of Energy (DOE) Office of Science User Facility operated for the DOE Office of Science by Argonne National Laboratory under Contract No. DE-AC02-06CH11357.

## Author contributions

J.D.H. devised the experimental strategy, purified the protein, and solved the structure. B.C. carried out the thermal stability assays. B.C. and L.R. purified the protein. F.L. optimized the crystallization conditions and harvested crystals. C.G., R.E.M, and K.C. carried out the molecular biology and baculovirus expression of GPR40. J.P.L. synthesized compound **1**. J.B. collected and processed X-ray diffraction data. S.D.K. and A.P.L. performed the radioligand binding assays. K.L.W. and K.A.O. performed the functional assays. Y.C., M.S., J.P.R. H.-C.Y., and N.A.R. performed the OGTT, IPGTT and incretin secretion assays. T.G. performed the analytical fluorescence size-exclusion chromatography. T.G. and M.D.S. developed the mass spectrometry protocol for analyzing the GPR40 crystallization samples. M.R.L. performed and analyzed the computation modeling of AM-1638, TAK-875, stearic acid and γ-LA. Supervision of the work was carried out by J.D.H, C.M.-R., A.R.M., and C.H. The manuscript was prepared by J.D.H., B.C., L.R., S.D.K., K.L.W., K.A.O., M.R.L., J.P.L., Y.C., J.B., K.A.B., A.R.M., and C.H. All authors contributed to the final editing and approval of the manuscript.

## Additional information

**Competing interests:** All authors are current or past employees of Eli Lilly and Company. The authors declare no competing financial interests.

