## [Peer Review File · Nature Communications]

REVIEWERS' COMMENTS:

Reviewer #2 (Remarks to the Author):

This is a paper previously reviewed by me elsewhere. I have reviewed the original concerns of mine and the authors have adequately addressed these. I recommend publication without delay.

Reviewer #3 (Remarks to the Author):

I am overall satisfied with the response to my previous concerns.

Reviewer #4 (Remarks to the Author):

The manuscript has been improved considerably. The new aspects of the overall complex allosteric story is very interesting and probably 'as good as it gets'.

Reviewer #5 (Remarks to the Author):

In their study, Ho and colleagues investigate how binding distinct allosteric sites of the same receptor generates biased G-protein signaling at the GPR40 receptor. Because this receptor is emerging as a new therapeutic target in diabetes, the authors also study the effects of compound 1 on insulin secretion and glucose metabolism. The data show that this compound strongly increases insulin and incretin plasma levels and that it improves glucose tolerance dose dependently. Contrary to the authors' claim, however, the results do not demonstrate unequivocally that the compound directly stimulates insulin secretion, as stated in the abstract. Even using IPGTTs, the in vivo effects on insulin secretion are confounded by many variables. There are two possibilities here. The authors could either rephrase their conclusion in the abstract by simply stating that compound 1 improves glucose metabolism and increases insulin plasma levels, while acknowledging that they don't know where it acts (i.e. it has direct and indirect effects on insulin secretion). This could reduce the impact of the paper. Or they could test if the compound has a direct effect on insulin secretion by performing secretion studies on isolated islets. These studies are simple and straightforward and are performed routinely at the Lilly facilities in Indianapolis.

Having said this there were some issues related to the results in Figure 1:

- 1) Insulin secretion in Figure 1d is almost absent from mice treated with vehicle. It is unclear why there is no insulin response to the glucose challenge in this control group.
- 2) It is unclear why different mouse strains (C57B6 and ICR) are used in the different glucose tolerance tests. There are large differences in glucose metabolism between mouse strains. Thus, the use of different mouse strains needs to be justified.
- 3) Along these lines, what is the genetic background of the GPR40 KO mice? Also, insulin secretion in the KO mice is not different from that shown for the control in Figure 1d, probably because insulin secretion is absent in the control.

Response to review #5 (final round of review) NCOMMS-17-11997-A:

We would like to thank all the reviewers for their excellent critiques as they have made our manuscript stronger. Please find our responses to Reviewer #5 below in blue.

Reviewer #5 (Remarks to the Author):

In their study, Ho and colleagues investigate how binding distinct allosteric sites of the same receptor generates biased G-protein signaling at the GPR40 receptor. Because this receptor is emerging as a new therapeutic target in diabetes, the authors also study the effects of compound 1 on insulin secretion and glucose metabolism. The data show that this compound strongly increases insulin and incretin plasma levels and that it improves glucose tolerance dose dependently. Contrary to the authors' claim, however, the results do not demonstrate unequivocally that the compound directly stimulates insulin secretion, as stated in the abstract. Even using IPGTTs, the in vivo effects on insulin secretion are confounded by many variables. There are two possibilities here. The authors could either rephrase their conclusion in the abstract by simply stating that compound 1 improves glucose metabolism and increases insulin plasma levels, while acknowledging that they don't know where it acts (i.e. it has direct and indirect effects on insulin secretion). This could reduce the impact of the paper. Or they could test if the compound has a direct effect on insulin secretion by performing secretion studies on isolated islets. These studies are simple and straightforward and are performed routinely at the Lilly facilities in Indianapolis.

In regard to the reviewer's request for insulin secretion data in islets, we believe that the absence of increased insulin levels by compound 1 in GPR40 KO mice provides sufficient evidence that the insulin response is GPR40-mediated. The scope of our manuscript is not intended to address the extent of the direct or indirect effect of compound 1 on insulin secretion. Our study demonstrates the effect of compound 1 on plasma insulin and glucose is GPR40 mediated regardless of site of action. To improve clarity, we have re-written the pharmacology section of compound 1. It now reads this way in lines 77-86:

Compound 1 is a full agonist (relative to the native ligand) that acts to increase intracellular Ca²⁺ and cAMP levels as a result of GPR40 coupling to both Gα_q and Gα_s (Fig. 1b-c). In intraperitoneal glucose tolerance test (IPGTT), compound 1 dose-dependently increases insulin plasma levels and improves glucose metabolism (Fig. 2a-d). Absence of increased insulin levels by compound 1 in GPR40 KO mice suggests the insulin secretion is GPR40-mediated (Fig. 2c). Compound 1 also exhibits similar dose-dependent reduction of blood glucose in oral glucose tolerance test (OGTT) that is GPR40-mediated (Fig. 2e-g). In addition, oral administration of compound 1 demonstrates robust increase in incretins

(GLP-1 and GIP) that is GPR40-mediated (Fig. 3a-d). Compound 1 does not stimulate peptide YY (PYY) release in vivo (Fig. 3e) suggesting that increased incretin secretion is mechanism mediated instead of degranulation.

Having said this there were some issues related to the results in Figure 1:

1) Insulin secretion in Figure 1d is almost absent from mice treated with vehicle. It is unclear why there is no insulin response to the glucose challenge in this control group.

Insulin secretion in response to the glucose challenge is minimal in this model. It's not obviously visible in the scale of insulin levels stimulated by compound 1. (Pharma Res Per, 4(6), 2016, e00278, doi: 10.1002/prp2.278)

2) It is unclear why different mouse strains (C57B6 and ICR) are used in the different glucose tolerance tests. There are large differences in glucose metabolism between mouse strains. Thus, the use of different mouse strains needs to be justified.

Using ICR in screening IPGTT assay is for a practical reason. ICR mice cost significantly less compared with C57B6.

3) Along these lines, what is the genetic background of the GPR40 KO mice? Also, insulin secretion in the KO mice is not different from that shown for the control in Figure 1d, probably because insulin secretion is absent in the control.

The genetic background of the GPR40 KO mice is C57B6. Insulin secretion in response to the glucose challenge is very minimal in C57/B6 mice (unpublished data). The key message from this study is diminishing of robust insulin secretion by compound 1 in KO mice demonstrating the effect is GPR40 mediated.